# Assessment of Retinal Pigment Epithelium Alterations and Chorioretinal Vascular Network Analyses in Patients under Treatment with BRAF/MEK Inhibitor for Different Malignancies: A Pilot Study

**DOI:** 10.3390/jcm12031214

**Published:** 2023-02-03

**Authors:** Giuseppe Fasolino, Gil Awada, Laura Moschetta, Jorgos Socrates Koulalis, Bart Neyns, Bert Verhelst, Peter Van Elderen, Pieter Nelis, Paul Cardon de Lichtbuer, Wilfried Cools, Marcellinus Ten Tusscher

**Affiliations:** Ophthalmology Department, Universitair Ziekenhuis Brussel, 1090 Brussels, Belgium

**Keywords:** BRAF/MEK inhibitor adverse effect, central serous chorioretinopathy, MEK-inhibitor-associated retinopathy

## Abstract

In the last two decades, an increasing number of so-called molecular-targeted therapies have become available for the treatment of patients with advanced malignancies. These drugs have included inhibitors of proteins in the MAPK pathway, such as BRAF and MEK inhibitors, which are characterized by a distinct toxicity profile. The eye is particularly susceptible to adverse effects due to MEK inhibitors, and the term MEKAR (MEK-inhibitor-associated retinopathy) indicates the presence of subretinal fluid, mimicking central serous chorioretinopathy (CSC). The pathogenesis of the retinal alterations related to MAPK pathway inhibitors is still unclear, and questions are still open. The present study aims to assess the presence of retinal pigment epithelium alterations as predictive parameters for retinal toxicity, analyzing, at the same time, the chorioretinal vascular network in patients undergoing BRAF/MEK inhibitor treatment for different malignancies.

## 1. Introduction

In the last two decades, an increasing number of so-called molecular-targeted therapies have become available for the treatment of patients with advanced malignancies. These drugs, which can be administered orally or intravenously, act by specifically binding target proteins in or on the cancer cells or in the tumor microenvironment, which directly or indirectly leads to the suppression of tumor cell growth. These targets are generally mutated and constitutively activated oncogenic proteins but can also be normal signaling proteins that are hyperactivated by upstream stimuli. Molecular-targeted therapies can block cellular receptors, intracellular signals, or circulating factors. Some molecular-targeted therapies have a high affinity for a single target (for example, dabrafenib blocks mutant BRAF^V600^), while others have more pleiotropic activity (for example, the oral multitargeted kinase inhibitor, regorafenib, blocks mutant BRAF^V600^, wild-type BRAF and CRAF, and VEGF receptors, among others) [1]. BRAF and MEK inhibitors act by blocking the mutant BRAFV600 protein and MEK protein (mitogen-activated protein kinase), respectively, in the mitogen-activated protein kinase pathway (MAPK). MAPK is a signaling pathway that is frequently constitutively activated in cancer (through upstream activation or mutations in the signaling proteins) and leads to cancer cell proliferation, survival, and metastasis [2]. In patients with advanced BRAF^V600^ mutant melanoma, the combination of a BRAF inhibitor, which blocks mutant BRAF^V600^, and an MEK inhibitor, which blocks the downstream protein MEK, has shown impressive activity, with objective response rates ranging from 64 to 70% [3,4,5]. Adverse events of molecular-targeted therapies are caused by the off-target blockade of proteins in healthy tissues. An example includes the skin rash caused by EGFR inhibitors by the blockade of EGFR in keratinocytes [6].

Particular toxicity of MAPK pathway inhibitors, in particular, MEK inhibitors, involves the eye, with an incidence of up to 90%. The term MEKAR (MEK-inhibitor-associated retinopathy) includes the class effect dose/time-dependent retinal adverse events observed with the use of MEK inhibitors. The clinical characteristics of MEKAR involve blurred vision, transient visual disturbances, flashes, and subretinal fluid, mimicking central serous chorioretinopathy (CSCR). Most of these adverse events are a- or paucisymptomatic, self-limited, or reversible with temporary drug withdrawal or dose decrease [7]. The proposed pathophysiologic mechanism to explain the presence of subretinal fluid involves the off-target blockade of MEK (and other signaling proteins in the MAPK pathway) in retinal pigment cells, where the MAPK pathway is responsible for maintaining the integrity of the retinal pigment epithelium, leading to hyperpermeability and disruption of the blood–retinal barrier [8]. Regarding this hypothesis, some authors describe an abnormal electrooculography exam, combined with a high incidence of MEKAR, in patients under treatment with an MEK inhibitor (binimetinib) for metastatic cutaneous and uveal melanoma [9]. Nevertheless, the pathogenesis of MEK-inhibitor-associated retinopathy is still unclear, and questions are still open about the possible toxicity related to different new types of MEK inhibitors, used alone or in combination with BRAF inhibitors, or on the possible involvement of the chorioretinal vascular network in this process, as we observe during CSCR.

Optical coherence tomography angiography (OCT-A) is a recently developed non-invasive imaging technique, which extrapolates the change in the OCT signal that originates from the flow of blood cells, being able to map the microvasculature of the retina and choroid [10]. So far, OCT-A has been employed to research both qualitative and quantitative microvascular data in various ocular diseases. [10]

This study aims to assess the possible interaction between BRAF/MEK inhibitors (dabrafenib–trametinib) and retinal pigment epithelium as predictive parameters for retinal toxicity, analyzing, at the same time, the chorioretinal vascular network by OCT-A.

## 2. Materials and Methods

### 2.1. Patient Characteristics and Study Design

In this single-center study, we identified and included 17 eyes of 9 adult patients with advanced (unresectable or metastatic) cancers who were treated with oral therapies targeting the MAPK pathway. These drugs included BRAF inhibitors (dabrafenib) and MEK inhibitors (trametinib). They could be employed as a monotherapy or in combination with other MAPK pathway inhibitors. Dosing of the MAPK pathway inhibitors was at the discretion of the referring oncologist, according to the drug label (in case of approved indications) or according to the study protocol (in case of inclusion in a clinical trial). Patients with CAR (cancer-associated retinopathy) and AIR (autoimmune retinopathy) were excluded. The patients were examined in the Department of Ophthalmology of the University Hospital of Brussels. The aim of this study was to complete a mean follow-up period of 3 months, with an ophthalmologic check-up every 2 weeks for the 1st month and every 4 weeks for the other 2 months. Considering that the flow void area is increased in eyes with high myopia, subjects with refraction greater than −6 diopters were excluded. [11] Further exclusion criteria were subjects with uncontrolled glaucoma or uncontrolled diabetes mellitus, uveal melanoma, history of uveitis of other retinal diseases, history of retinal laser or photodynamic therapy, history of uncontrolled hypertension, history of hypercoagulability of hyperviscosity syndromes. Before initiation of treatment, and during each follow-up visit, patients underwent an ophthalmologic exam, including a thorough history, automated refraction, best-corrected visual acuity (BCVA) measurement, biomicroscopy, fundus autofluorescence (FAF), fundus pictures, fundus examination, OCT, and OCT-A. In addition, electrooculography (EOG) was performed according to the guidelines of the International Society for Clinical Electrophysiology of Vision [12]. In the patients with MEK-inhibitor-associated retinopathy (MEKAR), the check-up was supplemented with fluorescein angiography (FA). This single-center, prospective cohort study complies with the Declaration of Helsinki and was approved by the ethical committee of the University Hospital of Brussels. All included patients signed for written informed consent. Table 1 features a complete list of patient characteristics.

### 2.2. Baseline Characteristics

A total of 9 patients (17 eyes) were included in this analysis. The clinical baseline characteristics are included in Table 1.

### 2.3. Image Acquisition and Evaluation

SD-OCT and OCT-A images were acquired using RTVue XR Avanti. A 6 × 6 mm HD OCT-A scan, centered around the macula, was obtained for every patient. Segmentation of the different layers (superficial capillary plexus, deep capillary plexus, outer retina, and choriocapillaris) was accomplished using the built-in software segmentation algorithm [13,14]. The data were analyzed using the open-source ImageJ expansion, Fiji [15,16]. To quantify the data, they were first turned into an 8-bit image and binarized using the Phansalkar method [17,18]. Vessel area density of the superior and deep capillary plexus was obtained using the “analyze particles” command. For calculating the number and total area of the flow voids, the image size was adjusted to the inner 3 × 3 mm image, binarized using the Phansalkar Method, despeckled, and, lastly, analyzed using a particle size of 10.000 µm^2^ or greater [19,20]. OCT-A 3 × 3 mm could be considered as a technical limitation in place of 6 × 6 mm, but we chose this parameter to guarantee fewer image artifacts.

### 2.4. Statistical Analysis

Statistics were performed using R for SPSS Statistics (version 4.1.3). Paired sample *t*-test and Wilcoxon signed-rank test were used as indicated. A general linear mixed model was used for equivalence testing and for estimating the average drop and recovery in the light peak:dark trough ratio after the initiation of treatment. Statistical significance was assumed by *p*-values < 0.05. The AIC (Akaike information criterion) was used to estimate the interaction between dosage (low/high) and time.

## 3. Results

Statistical analysis was performed on 15 eyes of 8 patients. Of the nine patients originally included, one patient dropped out because of rapid tumor progression. At baseline, visual acuity was between 0.8 and 1.0, and it remained stable during the follow-up visits, including the patient that developed MEKAR. Of the eight treated patients that completed the follow-up, one patient presented with the clinical characteristics of MEKAR in one eye; the other eye had been enucleated in 2018 for the presence of invasive conjunctival melanoma. The patient developed subretinal fluid (SRF) and multiple zones of retinal pigment epithelium (RPE) detachment during the first two weeks after the initiation of the treatment (trametinib 2 mg daily plus dabrafenib 50 mg twice daily) without symptoms (Figure 1). At the same time, the EOG light peak:dark trough ratio dropped from 3.5 at baseline to 1.8 at the first control, representing a single case with a loss of 49%, expressing possible acute damage in the RPE, thereby resulting in abnormal fluid and ion transport. In the next two weeks, the fluid reabsorbed spontaneously without any modification in the therapy (Figure 1 and Figure 2). Moreover, in this patient, we performed fluorescein angiography, which confirmed the presence of subretinal fluid inside the macula but without any sign of leakage that could show anatomical damage to the RPE (Figure 1). Autofluorescence showed no alteration.

The quantitative assessment of the OCT-A choriocapillaris slabs before and during treatment with trametinib/dabrafenib is illustrated in Table 2 and Table 3. A comparison revealed no significant difference in flow void number and total flow void area (mm^2^ and %) (*p* = 0.798; 0.308 and 0.308, respectively).

An equivalence test as part of a general linear mixed model failed to show equality in flow void numbers and total flow void numbers, with fairly large standard errors due to the limited number of eyes and patients.

An assessment of SCP and DCP vascular parameters is illustrated in Table 4 and Table 5. There was no significant difference between the VAD (vessel area density) of the SCP and DCP before and during treatment with trametinib (*p* = 0.293 and 0.582, respectively).

An equivalence test as part of a general linear mixed model failed to show equality in the SCP VAD and DCP VAD, with fairly large standard errors due to the limited number of eyes and patients.

Electrooculography (EOG) was performed according to the guidelines of the International Society for Clinical Electrophysiology of Vision. Before initiation of the oncological treatment, all the patients showed a normal light peak:dark trough ratio (≥1.8). During the first follow-up visit, two weeks after the initiation of treatment, EOG showed a subnormal light peak:dark trough ratio (<1.5–1.8>) in 11 eyes of 6 patients, and only 4 eyes of 2 patients maintaine50mg twice daily) without symptoms d a normal value (≥1.8) but with a decrease of 30% from the baseline.

A visual representation of the light peak:dark trough ratio is depicted in Figure 3. A general linear mixed model showed a dramatic decrease in the light peak:dark trough ratio after the initiation of oncological treatment (−0.76, confidence interval [−1.046; −0.481]; *p* = 0.001), followed by a suggested increase of 0.268 (confidence interval [−0.040; 0.577]; *p* = 0.105), which was shown visually, but with only a few patients, it was not significant, and finally, there was no indication of anything except consolidation (confidence interval [−0.041; 0.038]; *p* = 0.999). The data do not suggest any interactions between dosage (low/high) and time, as modeled (Figure 4 and Figure 5), based on the AIC (Akaike information criterion), when these interactions are included. The AIC was calculated with and without taking dabrafenib dosage into account at 89.9395 and 79.14193, respectively. The lower value in the latter group suggests that there is not enough evidence to say that the change in the light peak:dark trough ratio is dependent on dosage.

The low-dose group comprises nine eyes from five patients. The high-dose group comprises six eyes of three patients. The patient who developed a MEKAR belonged to the low-dose group. The error bars represent the standard deviation.

## 4. Discussion

In the present study, we report the temporal characteristics of the changes in the EOG exam, together with analysis by OCT-A of retinal and choriocapillaris circulation, in patients who underwent BRAF/MEK inhibitor treatment for different malignancies.

The results must be viewed as a hypothesis-generating pilot study and must be interpreted with caution because of the small sample sizes. Further studies are needed to confirm our findings in a large population and other factors can be involved in retinal toxicity related to oncological target therapies. Moreover, we cannot exclude the presence of scleral and vortex vein drainage alterations as causes of subretinal fluid.

To the best of our knowledge, only one previous study (from our group) [21], using a different cohort of patients, evaluated the use of OCTA to investigate the clinical effects of oncological target therapy on retinal vascularization. Nevertheless, in this research, we focused our attention on retinal pigment epithelium alterations, which are detectable during treatment with BRAF/MEK inhibitors, as predictive parameters for the development of retinal toxicity.

The RPE fulfills multiple roles that are essential for visual function: (1) protecting the macula from the light that is focused by the lens; (2) feeding photoreceptors with substances, such as glucose and vitamin A, present in the blood; (3) transporting water present in the subretinal space to the blood through the transepithelial CL- pump; (4–5) maintaining visual function and the visual cycle; (6) the phagocytosis of the photoreceptor outer segment; (7) secretion of a large variety of factors and signaling molecules, such as vascular endothelial growth factor (VEGF) and pigmented epithelium-derived factor (PEDF); and (8) maintaining immune privilege in the eye [22,23].

Multiple factors are involved in RPE functions, and different ion channels and transporters regulate the cell volume and maintain water osmosis during the visual cycle. The electrooculogram (EOG) reflects the ion conductance of the RPE. The index known as the “light rise” or “light peak” substance, which is released by the photoreceptors and whose chemical makeup is unknown and interacts either directly or indirectly with the RPE, is what causes the light rise in the EOG. There is indirect evidence that this chemical increases the calcium concentration inside of the RPE’s cells ([Ca2+]in), which in turn activates a basolateral ionic Cl- channel, depolarizing the basal membrane and producing the EOG’s recognizable light rise. The light peak:dark trough ratio is the common way to express this as a ratio of “light peak to dark trough.” The standing potential of the entire RPE is altered by an aberrant EOG, and the RPE’s ability to pump fluid out of the sub-retinal region is disrupted, leading to the buildup of serous SRF [24,25,26,27].

Previous studies have described the retinal lesions associated with MEK inhibition treatment as central serous-like chorioretinopathy (CSCR) [28,29]. It has been demonstrated that CSCR has a higher total choriocapillaris flow void area and number of flow void lesions ≥10,000 μm than healthy eyes [30]. There was no discernible change in total flow void area or flow void number (≥10,000 μm) between OCT-A pre- and post-initiation in this investigation, as well as in our prior one. No differences were found in a similar examination of flow characteristics in the superficial and deep plexus. Instead, we found that the most striking abnormality was the profoundly abnormal EOG, expressed by a dramatic decrease in the light peak:dark trough ratio, confirming what other authors have observed in patients treated only with MEK inhibitors (binimetinib) [9]. This suggests that the main issue is panretinal RPE dysfunction, which can result in serous retinopathy caused by the RPE pump failing. Considering the relationship between the beginning of the treatment and the rapid onset of the abnormal EOG, we assume that the RPE alteration is related to a direct interaction with the BRAF/MEK inhibitors through the MAPK pathway. In fact, altering this channel may impair typical fluid flow and cause fluid to collect under the retina. The fluid transport channel, aquaporin 1, which has been shown to be controlled by the MAPK pathway, is a component of this mechanism. The effective transepithelial water transport across the RPE is likely facilitated by AQP1 in the RPE in vivo, which also helps to maintain retinal attachment and avoid subretinal edema [31,32] (Figure 4).

Otherwise, we observed a low incidence (one eye) of MEKAR in our patient group, as compared to other cohorts, and no other types of ocular adverse events, as reported in the literature [33]. As described above, the MAPK pathway appears to be primordial for the function of the RPE, and previous reports have shown that blockade by MEK inhibitor monotherapy can lead to retinal abnormalities such as MEKAR. In non-malignant BRAF wild-type cells (such as retinal pigment epithelial cells), treatment with BRAF inhibitors (for example, dabrafenib) can cause the phenomenon of paradoxical MAPK pathway activation by the transactivation of wild-type RAF [34]. This paradoxical activation will lead to some degree of activation of the MAPK cascade despite downstream MEK inhibition and might explain the incidence of the subclinical retinal abnormalities that we report.

Another explanation for the failure of the retinal cells is that an autoantibody attack against certain RPE epitopes may compromise the RPE pumping function, with a consequent alteration of the EOG. Paraneoplastic retinopathies (PRs), such as cancer-associated retinopathy (CAR) or melanoma-associated retinopathy (MAR), represent retinal disorders mediated by autoimmune mechanisms and are associated with serum anti-retinal autoantibodies [35].

Sawyer was the first to describe CAR [36]. The etiology is connected to an antibody synthesis against Recoverin, a calcium-binding protein found in retinal photoreceptors, and it is typically associated with small-cell lung cancer [37]. Rod and cone dysfunction can cause bilateral vision loss over several months, and in 50% of cases, visual symptoms are present before a diagnosis of systemic cancer.

MAR was described for the first time by Berson and Lessell [38]. This is related to the common neuroectodermal origin of the melanocytes and retinal cells [39]. According to Keltner et al. [40], there is a latent period of ~3.6 years between the diagnosis of the primary neoplasm and the onset of MAR.

Furthermore, the presence of anti-RPE antibodies against the RPE protein, bestrophin, was described in a patient with metastatic choroidal melanoma and vitelliform paraneoplastic retinopathy, characterized by an abnormal EOG [41].

There are few data on the potential prevalence of these autoantibodies in individuals, and there is currently no gold standard for the diagnosis.

In patients with presumed PR, the presence of autoantibodies can predict the presence of an underlying neoplasm (i.e., anti-recoverin autoantibodies), but their diagnostic value is not fully understood [42,43,44].

Considering the rapid onset of the profoundly abnormal EOG and the dramatic decrease in the light peak:dark trough ratio in all populations, we hypothesize that panretinal dysfunction of the RPE can be related to the direct toxicity of the oncological treatment (by the blockade of the MAPK pathway) and, consequently, represents an important predictive parameter for retinal alteration. The low incidence of serous retinopathy observed in our population compared with other studies could be related to the phenomenon of paradoxical MAPK pathway activation caused by BRAF inhibitors. The unremarkable OCT-A analysis suggests no role for the retinal and choriocapillaris vasculature.

## Figures and Tables

**Figure 1 jcm-12-01214-f001:**
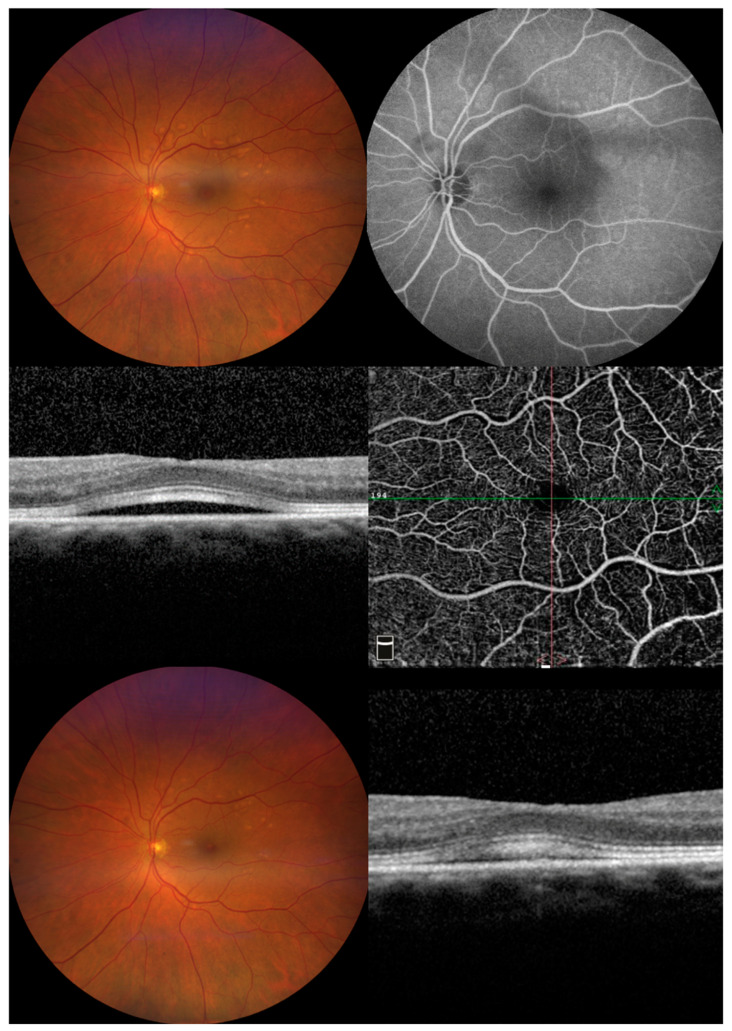
Images from patient number 6 affected by MEKAR. (**Upper left**) At two weeks from the start of treatment, the color picture shows multiple retinal pigment epithelial detachments in the posterior pole. (**Upper right**) FAG shows no leakage in the macular area; the presence of pooling corresponds to localized pigment epithelial detachment. CSCR was ruled out. (**Middle left**) Macular optical coherence tomography shows foveal serous neuro-epithelial detachment with thickening and high reflectivity of interdigitation zone. No sign of increased choroidal thickness or dilatation of choroidal vessels can be seen. (**Middle right**) Enface OCTA images of the superficial vascular plexus segmentation show no visible alteration. (**Bottom left**) At four weeks from the start of treatment, the color picture shows partial reabsorption of the multiple retinal pigment epithelial detachments in the posterior pole. (**Bottom right**) Macular optical coherence tomography shows a decrease in serous neuro-epithelial detachment.

**Figure 2 jcm-12-01214-f002:**
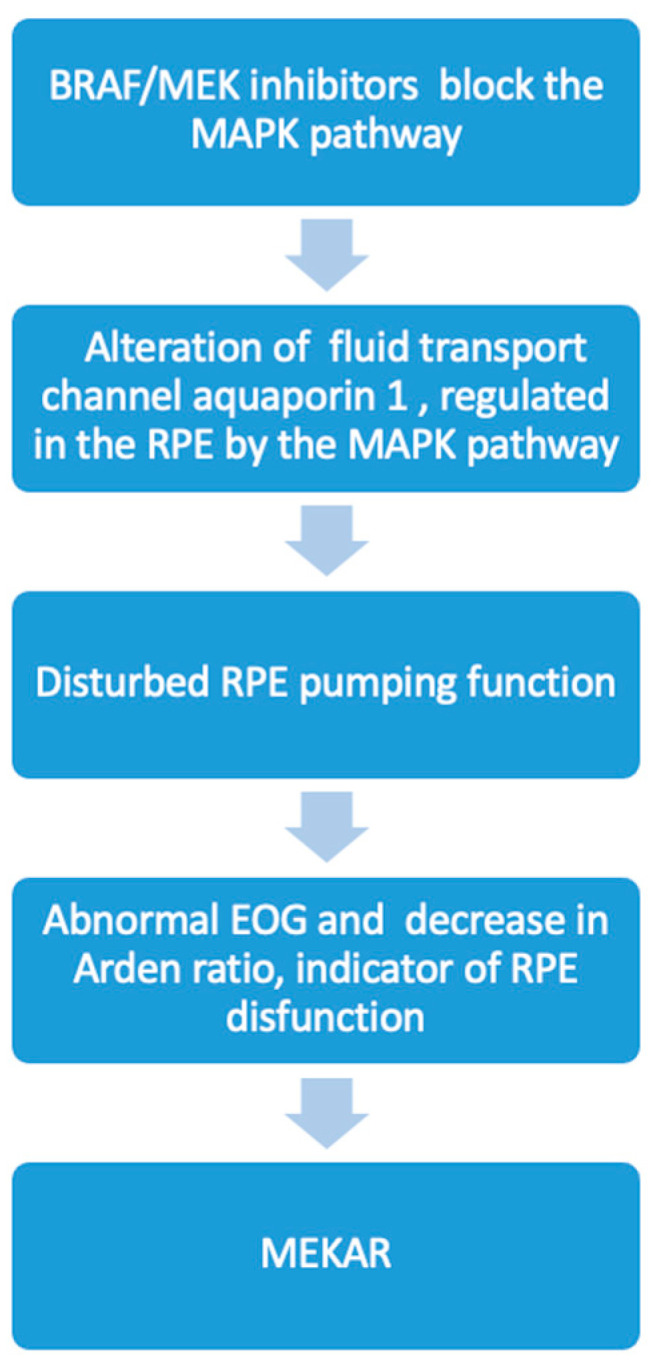
Flow chart of proposed interaction between BRAF/MEK inhibitors and RPE in MEKAR and its possible correlation with Arden ratio decrease.

**Figure 3 jcm-12-01214-f003:**
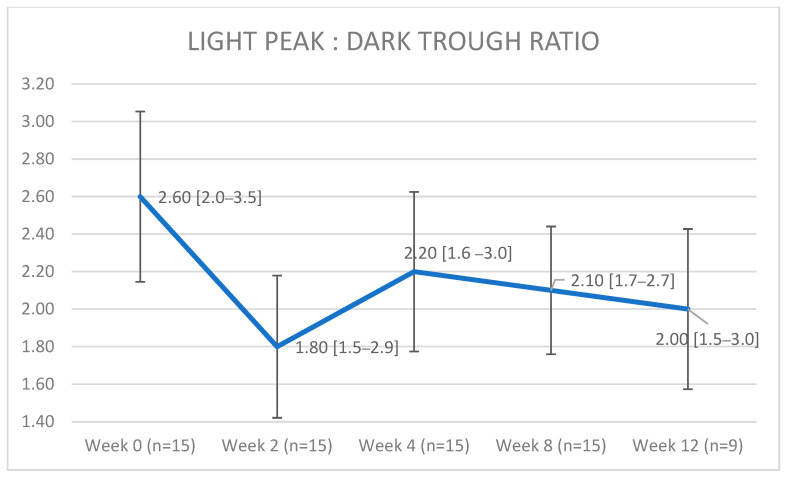
A visual representation of the light peak:dark trough ratios shows a dramatic decrease after the initiation of oncological treatment, followed by an increase, and, lastly, by suggested consolidation. Data are visualized as median (range); the error bars represent the standard deviation.

**Figure 4 jcm-12-01214-f004:**
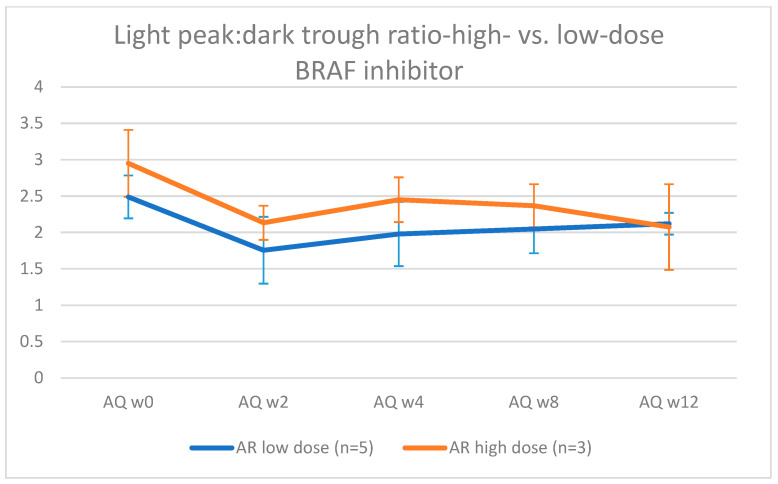
Visual representation of the evolution of light peak: dark trough ratio in eyes treated with a low (50 mg/day) vs. high dose (150 mg/day) of dabrafenib.

**Figure 5 jcm-12-01214-f005:**
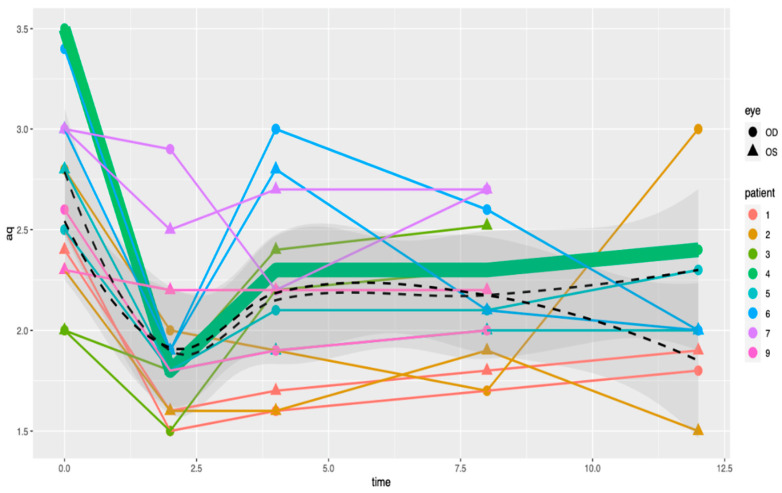
A visual representation of the light peak:dark trough ratios based on single patient, which does not show a decrease in the light peak:dark trough ratio regardless of the therapeutic dose.

**Table 1 jcm-12-01214-t001:** Clinical baseline characteristics.

Patient (Eyes)	Sex	Age	Cancer Type	Drug (Dosing)	Relevant Medical History
**1 (2)**	Male	56	MEL	Trametinib (2 mg QD) plus dabrafenib (50 mg BID)	None
**2 (2)**	Male	58	MEL	Trametinib (2 mg QD) plus dabrafenib (150 mg BID)	None
**3 (2)**	Male	59	MEL	Trametinib (2 mg QD) plus dabrafenib (50 mg BID)	None
**4 (2)**	Male	58	MEL	Trametinib (2 mg QD) plus dabrafenib (150 mg BID)	Arterial hypertension
**5 (2)**	Male	55	GBM	Trametinib (2 mg QD) plus dabrafenib (50 mg BID)	None
**6 (1)**	Female	65	MEL	Trametinib (2 mg QD) plus dabrafenib (50 mg BID)	Enucleation due to MEL
**7 (2)**	Female	45	MEL	Trametinib (2 mg QD) plus dabrafenib (50 mg BID)	None
**8 (2)**	Female	83	GLOM	Trametinib (1 mg OD QD) plus dabrafenib (150 mg BID)	Heart disease
**9 (2)**	Female	75	MEL	Trametinib (2 mg QD) plus dabrafenib (50 mg BID)	Hypertension

Abbreviations: BID—twice daily; QD—once daily; GBM—glioblastoma multiform; MEL—melanoma; GLOM—malignant glomus tumor.

**Table 2 jcm-12-01214-t002:** Choriocapillaris flow void parameters at each control.

	Week 0 (n = 15)	Week 2 (n = 15)	Week 4 (n = 11)	Week 8 (n = 9)	Week 12 (n = 6)
**Flow void number**	72.07 ± 17.68	71.67 ± 17.94	75.82 ± 12.88	72.44 ± 13.87	82.50 ± 4.46
**Total flow void area** (mm^2^)	2.46 ± 0.77	2.39 ± 0.89	2.10 ± 0.53	2.27 ± 0.75	1.93 ± 0.16
**Total flow void area** (%)	27.37 ± 8.63	26.55 ± 9.90	23.29 ± 5.92	25.19 ± 8.30	21.47 ± 1.83

**Table 3 jcm-12-01214-t003:** Comparison of choriocapillaris flow void parameters before initiation of treatment and during follow-up.

	Pre-MEK/BRAF Inhibitor	During Follow-Up *	*p*-Value
**Flow void number ^1^**	72.07 ± 17.68	72.33 ± 14.80	0.798
**Total flow void area ^2^** (mm^2^)	2.46 ± 0.77	2.34 ± 0.76	0.308
**Total flow void area ^2^** (%)	27.37 ± 8.63	26.03 ± 8.40	0.308

^1^ Wilcoxon signed-rank test in non-normal distribution. ^2^ Paired sample *t*-test in normal distribution. * The follow-up values were obtained by calculating the mean of the follow-up measurements.

**Table 4 jcm-12-01214-t004:** SCP and DCP VAD at each control.

	Week 0 (n = 15)	Week 2 (n = 15)	Week 4 (n = 11)	Week 8 (n = 9)	Week 12 (n = 6)
SCP VAD (%)	37.10 ± 3.64	36.29 ± 4.29	36.51 ± 4.14	36.56 ± 4.17	38.22 ± 2.05
DCP VAD (%)	36.96 ± 7.47	36.73 ± 7.61	36.79 ± 7.96	36.84 ± 6.75	42.93 ± 4.17

**Table 5 jcm-12-01214-t005:** Comparison of SCP and DCP VAD before initiation of treatment and during follow-up.

	Pre-MEK/BRAF Inhibitor	During Follow Up *	*p*-Value
SCP VAD (%) ^1^	37.10 ± 3.64	36.37 ± 3.77	0.293
DCP VAD (%) ^1^	36.96 ± 7.47	36.36 ± 6.87	0.582

^1^ Paired sample *t*-test in normal distribution. * The follow-up values were obtained by calculating the mean of the follow-up measurements.

## Data Availability

Not applicable.

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
