# Peer review of "Assessment of Retinal Pigment Epithelium Alterations and Chorioretinal Vascular Network Analyses in Patients under Treatment with BRAF/MEK Inhibitor for Different Malignancies: A Pilot Study"

_jcm, 2023, doi:10.3390/jcm12031214_

Round 1

Reviewer 1 Report (New Reviewer)

The paper „Assessment of retinal pigment epithelium alterations and chorioretinal vascular network analyses in patients under treatment with BRAF/MEK inhibitor for different malignancies: a pilot study.” Brings novelty and original approach in treatment of retinal pigment epithelium alterations with BRAF/MEK inhibitor and after minor corrections should be accepted for publication.

Throughout the text when the authors address to more than one figure or table or graph it should be written for example Figure 2 and 3 instead fig. 2,3; or Table 4 and 5 instead Table 4,5.

Fig. 4 should be part of the results together with adequate text that introduces this Figure and not part of the discussion. The discussion part regarding the findings in Fig.4, nevertheless may obtain unchanged.

Before conclusion the main limitations and advantages (benefits) from this study should be emphasized. For instance the small sample size as main limitation should be given here before the conclusion part.

Conclusion should be rewritten. The sentence “In conclusion the present pilot study aims to assess the presence of retinal pigment  epithelium alterations as predective parameter for retinal toxicity due to BRAF/MEK inhibitor treatment in different malignancies” is already part of the introduction as should be and here it should be deleted. The limitations and the advantages of the study should be given before the conclusion part and the conclusion remarks should be only about the findings in this particular study.   

Author Response

Dear reviewer 

Thank you for your kind reply and your comments, which we found very helpful in improving our article. 

-we changed in our text the references to figures, graphic and tables as you suggested

-figure 4 stands for a visual representation of physio pathological hypothesis of MEKAR. It is based on the results as you have noticed. Nevertheless, it is not representative of pure objective results but instead it is enriched of diagnostic hypothesis and commentary. For this reason, we decided to maintain it in the discussion but thanking for your clarification we have changed the legend to clarify its aim

-we agree that the limitation and advantages of our study should be declared at the begin of discussion. We have made the suggested changes

-we improved our conclusion as requested

Reviewer 2 Report (New Reviewer)

Overall the paper is well-written and presented.

Although this prospective study is small, it supports an effect of such medications on the RPE as measured via EOG thereby potentially explaining the mechanisms of MEKAR.

Although choriocapillaris flow void measurements where taken, there was no mention of choroidal thickness (except in their case of MEKAR). This would be useful to mention in the results as it is a measure of choroidal vascular changes in serous chorioretinal conditions.

The use fundus autofluoresence may also be useful in examining changes of the RPE. As the follow-up was 3 months, any changes or lack of changes might be useful in the discussion in the context of RPE dysfunction and MAPK pathway disruption

Serous retinal conditions may be caused by processes involving the retina, RPE and choroid, but also the scleral (transcleral hydrostatic/oncotic pressure) and vortex vein drainage. Although the authors have found significant changes on the EOG from week 0 to 2, they must mention that they cannot rule out multiple/other processes that may drive MEKAR.

Further explanation on their results of consolidation in the discussion would be useful to the readers

Author Response

Thank you for your targeted comments which give us the opportunity to improve our work.

- Following our experience in this pathology and recent literature data’s1, choroidal thickness is not a key-point parameter in MEKAR diagnosis. For this reason, we decided to not include it in our study.

  1. Cerquaglia A, Lupidi M, Chhablani J, Gujar R, Iaccheri B, Fiore T, Fruttini D, Ramundo A, Vupparaboina KK, Castellani L, Simonetti E, De Carolis L, Tiacci E, Falini B, Cagini C. Choroidal vasculature analysis in MEK inhibitor-associated retinopathy. Eur J Ophthalmol. 2022 Nov;32(6):3564-3573. doi: 10.1177/11206721221081471. Epub 2022 Feb 23. PMID: 35195471.

-The aim of our pilot study was to investigate early RPE functional abnormalities that usually are not visible with morphological examinations like fundus autofluorescence. In addition, the gold-standard accepted in ophthalmologist community remains electro-oculogram (EOG). However, fundus autofluoresence is a cornerstone of diagnosis in retinal diseases and in the evaluation of RPE. Even if there are some case reports where fundus autofluorescence were reported with limited alteration, we do not register any abnormalities in our clinical series.

-we totally agree with the limitation that you have raised up and you can find these limitations reported at the beginning of the discussion

-we have improved the discussion as requested

Reviewer 3 Report (New Reviewer)

Authors aimed to study RPE function in patients undergoing BRAF/MEK inhibitor therapy for various cancers. They used multimodal imaging including FAF, OCT-A and EOG. They found no significant difference in findings of OCT-A pre and post therapy but notable difference was discovered in EOG during the follow up.

Authors have elaborated on OCT-A and EOG but not on FAF findings.

The area analysed on OCT-A was 3 x 3mm. This should be mentioned as a drawback/limitation.

Statistical analysis was done for OCT-A but not for EOG.

In the methods authors should mention if they have excluded patients with CAR / AIR

Page 7, Line 215, 217: “imagine” to be corrected to image

Page 8, Line 221: “no dilated choroidal vessels even.”- please correct the sentence

Page 8: OCT-A findings can be added

Figures are not cited in the text. The diagnosis should be mentioned. Blocked fluorescence on FA should be explained. 

Author Response

Thank you for your accurate comments which allow us to make our article more understandable and appropriate.

-The aim of our pilot study was to investigate early RPE functional abnormalities that usually are not visible with morphological examinations like fundus autofluorescence. In addition, the gold-standard accepted in ophthalmologist community remains electro-oculogram (EOG). However, fundus autofluoresence is a cornerstone of diagnosis in retinal diseases and in the evaluation of RPE. Even if there are some case reports where fundus autofluorescence were reported with limited alteration, we do not register any abnormalities in our clinical series.

- OCT-A was 3 x 3mm : we mentioned it as limitation in our methods and we explained the reason for that

-For statistical analysis we asked the contribute of our statistical department. They elaborate two models of statistical analysis regarding EOG reported in Methods session:

“general linear mixed model was used for equivalence testing and for estimating the average drop and recovery in light peak : dark through ratio after initiation of treatment. Statistical significance was assumed by p-values < 0,05”

“The data did not suggest any interaction between dosage (low/high) and time as modeled (graph. 2 and, 3), based on the AIC (Akaike information criterion), when these interac-tions are included. The AIC was calculated with and without taking Dabrafenib dosage in account, 89.9395 and 79.14193 respectively”

-we added the mention that you have suggested (CAR / AIR) in the methods

-we improved the spelling from imagine to image

-we improved the sentences as you suggest

-We added OCT-A findings as picture 2

-please see “fig.” at lines 135, 139,142

-On FA we described a presence of pooling corresponding to localized pigment epithelial detachment but not blocked fluorescence.

Round 2

Reviewer 3 Report (New Reviewer)

Absence of findings on FAF should be mentioned in the results.

"Image" spelling has to be corrected throughout the manuscript.

Page 8, Line 227: "even" to be changed to "seen"

Figures should be cited in the text. The diagnosis made for the patient in Figure 3 should be mentioned. Was CSCR ruled out in this case?

In the discussion authors have mentioned: "the most striking abnormality was the profoundly abnormal EOG, expressed by a dramatic decrease of the light peak:dark trough ratio." Statistical significance of this can be commented in the methods / in discussion.

Author Response

Dear Reviewer

thank you for your scrupulous analysis that helps to improve our manuscript further.

-We mentioned absence of findings on FAF in the results (line 146) as requested

-"Image" spelling has been corrected throughout the manuscript.

-We changed Page 8, Line 227: "even" into "seen"

-Figures are cited in the text in result; please find "fig." with navigation session in word

-The diagnosis made for the patient in Figure 3 -"Patient affected by MEKAR"-  has been mentioned in the legend in all the three figures. After your suggestion, we recalled the more significant clinical aspects in the legends.

 CSCR was indeed ruled out and we added this information in the legend as you suggested.

- We commented the statistical significance of dramatic decrease of the light peak:dark trough ratio in the methods paragraph 2.4, there we added also some info about AIC (Akaike information criterion).

"A general linear mixed model was used for equivalence testing and for estimating the average drop and recovery in light peak : dark through ratio after initiation of treatment. Statistical significance was assumed by p-values < 0,05. The AIC (Akaike information criterion) was used to estimate the interaction between dosage (low/high) and time."

This manuscript is a resubmission of an earlier submission. The following is a list of the peer review reports and author responses from that submission.

Round 1

Reviewer 1 Report

The present study aims to assess the presence of retinal pigment epithelium alterations as predictive parameter for retinal toxicity, analyzing at the same time the chorioretinal vascular network. In the last twenty years number of molecular-targeted therapies for the treatment of patients with advanced malignancies has increased. These drugs included inhibitors of proteins in the MAPK-pathway as BRAF and MEK-inhibitors. This study includes 9 patients (17 eyes; 5 males, 4 females; age 45-83) undergoing BRAF/MEK inhibitor treatment for different malignancies. Only a few studies have evaluated the use of optical coherence tomography angiography (OCT-A) to investigate the clinical effects of oncological target therapy on retinal vascularization. Statistical analysis was performed on 15 eyes of 8 patients since 1 patient dropped out due to the rapid tumour progression. The authors selected suitable statistical analysis but the sample size is smaller. The structure of the study is also appropriate. Authors hypothesize that panretinal dysfunction of the retinal pigment epithelium can be related to the direct toxicity of the oncological treatment (by blockade of the MAPK-pathway), and consequently represents an important predictive parameter for retinal alteration. Further studies are needed to confirm the authors findings. This manuscript reviews 47 articles and provides sufficient background information about recent research related to the topic. The topic of this manuscript is up to date, attractive and well suited for the Journal of Clinical Medicine. The manuscript is well written and divided into four parts. The text is clear and easy to read.  For better understanding authors used 5 figures, 3 tables and 1 graph including detailed photographs. I highly appreciate OCT-A and fluorescein angiography images which the authors included in this manuscript. This aid the reader's understanding and quality of the paper. I suggest checking for some minor spelling mistakes and grammar errors. Otherwise, I  have no major concerns regarding this manuscript

Author Response

Dear colleague. Thank you for your enthusiastic feedback toward our work. We have made some changes and improvements to the article we hope these will also be received by you with equal positive reaction. 

Reviewer 2 Report

1. The term MEKAR (MEK inhibitor Associated Retinopathy) is used to describe the class effect dose-/time-dependent retinal adverse events observed with the use of MEK inhibitors. The patients you selected were in different dose of dabrafenib. It would be better if you display every arden ratio in a table to show whether there is the error caused by the high dose group.

2. MEKAR usually presents acutely within the first week of the first dose. There will be some errors in the evaluation from the second week.

3.As you mentioned in the manuscript, the small sample sizes can not guarantee the correctness of the outcome.

Author Response

Dear Review thank you for your punctual suggestions.

  1. The analysis based on single patient shows that the decrease of Arden ratio is not dose-dependent. We are glad to add this new graphic to our article.
  2. Concerning the follow-up timing we would like to share with you two relevant articles. In the review by Mendez-Martinez and al. it was suggested to apply three monthly visits for asymptomatic patients. This choice was based on a previously study by Duncan where the most vulnerable period for MEKAR has been shown to be in the first three months. For greater accuracy of electrophysiological data, we decided to check the patient every two weeks during the first month. However, every patient was advised to contact us in case of new visual symptoms at any time.

-Méndez-Martínez S, Calvo P, Ruiz-Morena O, et al. OCULAR ADVERSE EVENTS ASSOCIATED WITH MEK INHIBITORS. Retina;39(8):1435-1450.

-Duncan KE, Chang LY, Patronas M. MEK inhibitors: a new class of chemotherapeutic agents with ocular toxicity. Eye (Lond) 2015;29:1003–1012

  1. A pilot study is defined as “A small-scale test of the methods and procedures to be used on a larger scale” (Porta, Dictionary of Epidemiology, 5th edition, 2008). We are aware of the limitation of a pilot study for definition and inside our case and we reported these in the article in our conclusion session as you saw.

But we consider our result a relevant point for future study on larger population and it is actually the aim of a pilot study.

Reviewer 3 Report

The research on MEKAR deserves people's attention. However, the result of this research cannot meet our expectations and the disscussion cannot convince us.

Firstly , the expression of the result part is confusing. Why not list the vision results of patients during follow-up? Why take the average value of data from the other 3 follow up in table2 and table3 instead of showing OCT-A for each follow-up? Why is there only 9 eyes for week12 in graph1? The time of Figures 1, 2 and 3, 4 is also inaccurate

Secondly, maybe, the main purpose of the article is to assess the possible interaction between BRAF/MEK inhibitor and PRE as predictive parameter for retinal toxicity, but we have not seen a specific indicator.

Finally, in the discussion part, the reseaecher pays more attention to the possible mechanism of MEKAR, but fails to explain the research results and their relationship with the mechanism, which confuses the focus of the discussion

Author Response

Dear colleague,

 thank you for your useful remarks. We totally agree with that MEKAR is a relevant topic that deserve people. Indeed in collaboration with our oncological department we have published 5 articles in the last two years. 

  1. Regard your first suggestion we decided to improve our article adding VA (in results session), table OCT-A in the follow-up (in results session), and reviewing figures legend. At the last follow up at 12 weeks, all the patients attended the control but the results of EOG of three patients were not reliable due to worsening of their clinical condition. For this reason we decided to not include these data in our analysis

2.3. As you can see in the introduction we declare clearly that the aim of the study is indeed to assess the possible interaction between BRAF/MEK inhibitor (dabrafenib-trametinib) and retinal pigment epithelium as predictive parameter for retinal toxicity. Following the point that you raised, we added a flow chart and we modified the discussion. Please see the abstract below. We would like also to underline that the gold standard in studying RPE function is the electro-oculogram and its calculation expressed by the " Arden Ratio".

" Instead, we found that the most striking abnormality was the profoundly abnormal EOG, expressed by a dramatic decrease of the Arden ratio, confirming what other authors have observed in patients treated only with MEK inhibitor (binimetinib) [10]. This indicates that the primary abnormality is a panretinal dysfunction of the RPE, which can lead to a serous retinopathy supposedly due to failure of the RPE pump. Considering the relationship between the beginning of the treatment and the rapid onset of the abnormal EOG, we hypotisize that the RPE alteration is related to a direct interaction with the BRAF/MEK inhibitors through the MAPK-pathway. Indeed modulation of this pathway may disrupt normal fluid transport and lead to an accumulation of fluid under the retina. A factor that is involved in this process is the fluid transport channel aquaporin 1 that was specifically shown to be regulated by the MAPK pathway. AQP1 in RPE in vivo probably contributes to the efficient trans-epithelial water transport across RPE, maintains retinal attachment, and prevents subretinal edema"